# Prevention of Testicular Damage by Indole Derivative MMINA via Upregulated StAR and CatSper Channels with Coincident Suppression of Oxidative Stress and Inflammation: In Silico and In Vivo Validation

**DOI:** 10.3390/antiox11102063

**Published:** 2022-10-19

**Authors:** Tayyaba Afsar, Suhail Razak, Janeen H. Trembley, Khushbukhat Khan, Maria Shabbir, Ali Almajwal, Nawaf W. Alruwaili, Muhammad Umar Ijaz

**Affiliations:** 1Department of Community Health Sciences, College of Applied Medical Sciences, King Saud University, Riyadh 12372, Saudi Arabia; 2Minneapolis VA Health Care System Research Service, Minneapolis, MN 55455, USA; 3Department of Laboratory Medicine and Pathology, University of Minnesota, Minneapolis, MN 55455, USA; 4Masonic Cancer Center, University of Minnesota, Minneapolis, MN 55455, USA; 5Atta-ur-Rahman School of Applied Biosciences, National University of Sciences and Technology, Islamabad 44000, Pakistan; 6Department of Zoology, Wildlife, and Fisheries, University of Agriculture, Faisalabad 38040, Pakistan

**Keywords:** cisplatin, indole, testicular tissues, molecular interactome, molecular docking, antineoplastic agent, reproductive hormones

## Abstract

Cis-diamminedichloroplatinum (II) (CDDP) is a widely used antineoplastic agent with numerous associated side effects. We investigated the mechanisms of action of the indole derivative N’-(4-dimethylaminobenzylidene)-2-1-(4-(methylsulfinyl) benzylidene)-5-fluoro-2-methyl-1H-inden-3-yl) acetohydrazide (MMINA) to protect against CDDP-induced testicular damage. Five groups of rats (*n* = 7) were treated with saline, DMSO, CDDP, CDDP + MMINA, or MMINA. Reproductive hormones, antioxidant enzyme activity, histopathology, daily sperm production, and oxidative stress markers were examined. Western blot analysis was performed to access the expression of steroidogenic acute regulatory protein (StAR) and inflammatory biomarker expression in testis, while expression of calcium-dependent cation channel of sperm (CatSper) in epididymis was examined. The structural and dynamic molecular docking behavior of MMINA was analyzed using bioinformatics tools. The construction of molecular interactions was performed through KEGG, DAVID, and STRING databases. MMINA treatment reversed CDDP-induced nitric oxide (NO) and malondialdehyde (MDA) augmentation, while boosting the activity of glutathione peroxidase (GPx) and superoxide dismutase (SOD) in the epididymis and testicular tissues. CDDP treatment significantly lowered sperm count, sperm motility, and epididymis sperm count. Furthermore, CDDP reduced epithelial height and tubular diameter and increased luminal diameter with impaired spermatogenesis. MMINA rescued testicular damage caused by CDDP. MMINA rescued CDDP-induced reproductive dysfunctions by upregulating the expression of the CatSper protein, which plays an essential role in sperm motility, MMINA increased testosterone secretion and StAR protein expression. MMINA downregulated the expression of NF-κB, STAT-3, COX-2, and TNF-α. Hydrogen bonding and hydrophobic interactions were predicted between MMINA and 3β-HSD, CatSper, NF-κβ, and TNFα. Molecular interactome outcomes depicted the formation of one hydrogen bond and one hydrophobic interaction between 3β-HSD that contributed to its strong binding with MMINA. CatSper also made one hydrophobic interaction and one hydrogen bond with MMINA but with a lower binding affinity of -7.7 relative to 3β-HSD, whereas MMINA made one hydrogen bond with NF-κβ residue Lys37 and TNF-α reside His91 and two hydrogen bonds with Lys244 and Thr456 of STAT3. Our experimental and in silico results revealed that MMINA boosted the antioxidant defense mechanism, restored the levels of fertility hormones, and suppressed histomorphological alterations.

## 1. Introduction

Cis-diamminedichloroplatinum (II), cisplatin, (CDDP) is a standard antineoplastic drug against solid neoplasms, including lung, hematologic, ovarian, testicular, colorectal, and head and neck cancer [1]. CDDP usage engenders numerous side effects, such as renal toxicity, hepatotoxicity, cardiotoxicity, and gonadal toxicity [2,3]. CDDP treatment against testicular cancer disrupts spermatogenesis [4] and decreases sperm quality [5], which in turn leads to azoospermia [6] and transitory or enduring negative impact on male reproductive organs [7]. Fifty percent of male cancer patients receiving cisplatin-based chemotherapy develop long-term secondary infertility, and medical treatment to prevent spermatogenic failure after chemotherapy is not available [8].

Spermatogenesis is an elegant phenomenon that coordinates the assembly of testicular cells, such as peritubular cells, Sertoli cells, and Leydig cells [9]. Synthesis of nucleic acid in germ cells, particularly in spermatogonia, is hindered by CDDP usage on account of its alkylating abilities [10]. CDDP-induced impairment of Leydig cells resulted in reduced testosterone secretion [11]. Testosterone, as an important sex androgen, plays a critical role in the growth, reproduction, and maintenance of the function of vital organs. Steroidogenic acute regulatory protein (StAR) is the rate-limiting enzyme for testosterone biosynthesis, by transporting cholesterol from the outer membrane of mitochondria to its inner side [12]. The biosynthesis of androgen is completed in serial reactions, in which the P450 side-chain cleavage enzyme (CYP20A1) is the first-step enzyme in the synthesis of testosterone, converting cholesterol to pregnenolone [13], while 3β-HSD catalyzes the conversion of dehydroepiandrosterone to androstenedione in mitochondria; thereafter, the process of biosynthesis of testosterone is continued in Leydig cells [14]. CDDP can induce Leydig cell dysfunction and testicular steroidogenic synthesis disorders in animals, inhibiting testosterone production and causing infertility [15]. Furthermore, CDDP decreases sperm motility and activation [16,17]. CatSper is a voltage-gated calcium channel that is expressed in sperm, and it leads to hyperactivated motility and fertility in males. The sperm-specific CatSper channel controls the intracellular Ca^2+^ concentration (Ca^2+^) [18]. In most mammals, hyperactivated sperm motility depends on calcium influx into the sperm cytoplasm either from the extracellular space or release from intracellular organelles [19]. Therefore, the CatSper channel controls sperm motility and hyperactivation. CatSper1 and CatSper 2 are localized only in sperm and play a crucial role in fertility [20]. Thus, it is important to determine whether chemotherapeutic drugs, such as CDDP, affect the expression of CatSper genes.

Therefore, the mitigation of CDDP-associated side effects is vital to improving the quality of life and the efficacy of anticancer drug therapy [21]. However, the pathophysiological mechanisms related to CDDP-associated side effects remain unresolved. Besides direct DNA damage, CDDP treatment leads to apoptosis/necrosis, excessive reactive oxygen species (ROS), inflammation, and oxidative stress [22]. Various approaches have been considered to minimize or prevent CDDP-associated side effects, such as the use of cytoprotective agents, antioxidants [23,24], and various medicinal plants [25].

Non-steroidal antiinflammatory drugs (NSAIDs), which are used for pain relief and antiinflammation, decrease COX-1 levels or activity [26]. Indole is an imperative scaffold in the field of medicinal chemistry, and indole-ring compounds represent a class of beneficial molecules and are proposed to substitute for prevailing medications in the future. Studies revealed that indole derivatives have shown vital pharmacological activities, such as antiinflammatory, antipyretic, anticancer, and analgesia [27,28]. The Andersson group discovered the treatment of sexual dysfunction by using the indole ring-containing compound Yohimbine [29]. Varied combinations of fused heterocyclic structures result in novel polycyclic configurations with assorted physical, chemical, and biological properties. Synthesis of novel derivatives of NSAIDs has enhanced their safety profile with amplified antiinflammation and reduced ulcerogenic activities. Novel indole derivative, 2-(5-methoxy-2-methyl-1H-indol-3-yl)-N’-[(E)-(3-nitrophenyl) methylidene] acetohydrazide (MMINA) was reported to have highly significant antiinflammatory and antioxidant activities as well as analgesic properties [30]. In our previous investigation, we observed a promising chemoprotective effect of MMINA against CDDP-induced organ injury by modulating antioxidant stress and regulating the expression of various signal transduction pathways involved in inflammation, i.e., NF-κB, STAT-3, IL-1, COX-2, iNOS, and TNF-α, in the kidney, liver, brain, and heart [31].

We aimed to investigate the effects of MMINA against CDDP-induced testicular toxicity in rats. To study the mechanism of action, we investigated the effects of MMINA treatment on CDDP-induced alterations on the expression of StAR, 3β-HSD, CatSper1, and CatSper2 genes. Expression of inflammatory biomarkers i.e., STAT 3, NFкB p65, TNF α, and COX-2 was also determined. Furthermore, we have analyzed antioxidant and oxidative stress markers, reproductive hormones, daily sperm count, and motility. Histopathological studies were performed to study the changes at the morphological level. Additionally, molecular docking studies were also performed to depict the molecular targets of MMINA against cisplatin organ damage.

## 2. Methods

### 2.1. Ethics Statement

The study protocol was approved by the ethics committee in the Department of Zoology, Wildlife, and Fisheries, University of Agriculture, Faisalabad, Pakistan (CEE Council 86/6011). Studies reported in the manuscript fully meet the criteria for animal studies specified in the ACS ethical Guidelines.

### 2.2. Synthesis of MMINA

2-(5-Methoxy-2-methyl-1H-indole-3-yl)-N’-[(E)-(3-nitrophenyl) methylidene]acetohydrazide was synthesized by the department of pharmaceutical chemistry, College of Pharmacy, King Saud University 2457, Saudi Arabia. The detail of compound synthesis and biological characterization has been reported. The compound was fully characterized by the spectral data and details are published [31] and included in Appendix A. Compound (MMINA), with para dimethylaminophenyl substitution, was found to have the highest antioxidant activity. It was further evaluated in vivo for various analgesic, antiinflammatory, ulcerogenic, and COX-2 inhibitory activity in different animal models. Lead compound (MMINA) was found to be a significant antiinflammatory and analgesic agent. It was also evaluated for ulcerogenic activity and demonstrated a significant ulcerogenic reduction in the ethanol and indomethacin model [30,32].

### 2.3. Drug Preparation

Cisplatin/CDDP was purchased from Sigma-Aldrich, San Diego, CA, USA. A dosage of 12 mg/kg body weight was selected according to previous literature to induce tissue damage [33,34]. A 25 mg/kg body weight dose of MMINA was selected based on our previous work [31].

### 2.4. Acute Toxicity Testing of DMFM

Acute toxicity testing of MMINA was done previously [31] following the guidelines of the Organization for Economic Cooperation and Development (OECD) for testing chemicals for acute oral toxicity [35].

### 2.5. Animals

In total, 35 adult male Wistar rats weighing 220–240 g were obtained from the animal house of the Department of Zoology, Wildlife, and Fisheries, University of Agriculture, Faisalabad, Pakistan.

Animals were acclimatized for a week before the onset of experiments. Animals were housed under the following conditions: temperature 25 °C 12/12 h light and dark cycle, humidity 60 ± 10%, and a pathogen-free environment. The rats were fed a dietary formulation of protein (18.1%), fat (7.1%), carbohydrate (59.3%), and fiber (15.5%) with food and water being provided ad libitum.

### 2.6. Experimental Design and Drug Administration

The animals were divided into five groups, with seven rats per group:Group 1: Control group, received isotonic saline (i.p.);Group 2: DMSO group, received 1% DMSO (i.p.);Group 3: CDDP group, received a single dose of CDDP (12 mg/kg, i.p.) on day 8 of the experiment;Group 4: MMINA + CDDP group received MMINA (25 mg/kg, i.p.) daily dose for 2 weeks and a single dose of cisplatin (12 mg/kg, i.p.) on the 8th day of the experiment, 1 h after the dose of MMINA;Group 5: MMINA group, received MMINA (25 mg/kg, i.p.) for 14 days.

### 2.7. Sample Collection and Preparation

After 6 days of CDDP administration, the body weights of all animals were recorded before they were euthanized by injecting a ketamine/xylazine mixture (75/2.5 mg/kg, respectively, i.p.) [36]. Anesthetized rats were secured in a supine position and the blood was withdrawn via cardiac puncture and organ samples were taken from the testes. Blood was centrifuged for 15 min at 1200× *g* at 4 °C to separate serum and was stored at −70 °C. The testis tissues were quickly harvested. Part of one testis was fixed in 10% formalin for histopathological studies, and the other part of the testis was frozen with liquid nitrogen to be used for RNA extraction, enzyme analysis, and immunoblotting. The second testis and epididymis were processed for sperm parameter analysis.

### 2.8. Daily Sperm Count and Motility

Daily sperm production (DSP) in the testis was determined according to the method described previously [37]. No sperms in epididymis indicate the production efficiency of sperms. Briefly, the caput/corpus and cauda of the epididymis were isolated and weighed. The tissues were separately homogenized (caput and carpus together and cauda alone) in saline containing 0.5% triton × 100. The sperm mixture was diluted with a nigrosine stain containing saline. A 20 µL aliquot was pipetted to the Neubauer chamber and sperms were counted. The numbers of sperm in epididymis were obtained by multiplying with dilution factor and reported as per gram of the tissue. Sperm motility was evaluated immediately after dissection. The cauda epididymis was ripped using a scissor in 0.5 mL pre-warmed (37 °C) phosphate-buffered saline (pH 7.3), diluted with a drop of nigrosine stain. An aliquot of 50 µL was pipetted on a slide and sperm suspension was observed for 3–5 min under a microscope [38]. A total of 100 sperm/field were analyzed for motility by a technician blinded to the treatment groups. The process was repeated thrice and the average value was used as the total sperm motility.

The percentage of motile spermatozoa was determined by using the following formula:$$\mathrm{Percentage}\;\mathrm{of}\;\mathrm{Motile}\;\mathrm{spermatozoa}=\left[\frac{\mathrm{mean}\;\mathrm{number}\;\mathrm{of}\;\mathrm{motile}\;\mathrm{spermatozoa}}{\mathrm{total}\;\mathrm{number}\;\mathrm{of}\;\mathrm{spermatozoa}\;}\right]\;\times \;100\%$$

### 2.9. Hormone Analysis

The serum was used for the estimation of testosterone and Luteinizing hormone (LH) using CUSABIO (CatLogNo. CSB-E05100r and CSB-E12654r, respectively) ELISA kits according to the instruction of the manufacturer.

### 2.10. Antioxidant Status and Oxidative Stress Markers

Testicular tissue from different treatment groups was homogenized in ice-cold buffer (1 mM EDTA, 20 mM Tris-HCl, pH 7.5). All assays were performed following standard protocols. NO activity was measured by using My BioSource kits (Catalog No: MBS 480450). Afsar et al. methods were used to assess SOD and GPx activity [39,40]. The total antioxidant capacity of testicular tissues was analyzed by OxiSelect Total Antioxidant Capacity Assay (TAC assay, Cell Biolabs, Inc., San Diego, CA, USA) [17].

### 2.11. Gene Expression Analysis

Total RNA from the frozen testis tissue of each rat was extracted by using a kit, according to the manufacturer’s instructions (Promega Catalog No.# Z3101). The cDNA synthesis was performed using the Abi kit (Catlog#4368814). The reaction mixture was prepared to contain 10 µL FastStart Universal SYBR Green Master (Roche, Munich, Germany), 6 µM reverse primers, and 10 ul cDNA, with RNAse-free water added to a total volume of 20 µL. The amplification and real-time analysis were done for 40 cycles with the following conditions; 95 °C (10 min) to activate of FastStart Taq DNA polymerase; 60 °C (1 min) for amplification and real-time analysis. The gene expression levels were determined using 2^-ΔΔCT^. Primer sequences used are shown in Appendix A.

### 2.12. Extraction of Protein and Western Blot Analysis

Briefly, testis samples were lysed in RIPA buffer supplemented with freshly added protease and phosphatase inhibitor cocktail 1:100 (Sigma), and protein concentration was determined by a Bradford assay [41]. For SDS-Page, equal amounts of protein were loaded using BIO-RAD Mini protein TGX precast gels on the BIO-RAD Mini protein tetra system (10025025 REVA). After protein separation by gel electrophoresis, the proteins were transferred to PVDF membranes using BIO-RAD trans blot Turbo_™_ transfer system. The membranes (BIO-RAD, Hercules, CA, USA) were blocked for 2 h with either 5% BSA (Sigma) or 5% nonfat dry milk (BIO-RAD, Cat. #170-6404), then incubated with primary antibody overnight at 4 °C. The following antibodies were used for the assessment of inflammatory biomarkers: NFκB p65 (ThermoFisher Scientific Catalog No. # 2A12A7), STAT3 (Thermofisher Scientific Catalog No. # MA1-13042), COX2 (Invitrogen Cat. # PA1-9032), TNF-α (Abcam catalog No.# EPR19147), and β-actin (Abcam CatalogNo.#. ab49900). To study protein levels of StAR (ab233427, Abcam, Fremont, CA, USA), a mitochondrial fraction of testis was prepared [42], while a cytosolic fraction was used for expression analysis of 3β-HSD (PA5-103519, Thermofisher, Scientific Inc., Waltham, MA, USA). The homogenate was centrifuged at 600 rpm for 15 min to obtain the mitochondrial cloudy supernatant, which was separated carefully and transferred to a new tube, and centrifuged again at 12,000 rpm for 15 min. The resulting mitochondrial pellet was washed in sucrose buffer and recentrifuged at 12,000 rpm for 15 min. The pellet was re-suspended in sucrose buffer, and the protein content was measured by using the Bradford assay [43,44]. For examination of CatSper protein expression in epididymis sperms, the above procedure was followed using epididymis tissue lysate, and anti-CatSper antibodies (SC-33153; Santa Cruz Biotechnology Inc., Dallas, TX, USA) were used. The previously developed procedures with slight modifications were used to perform SDS-PAGE and Western blot investigations [43,45]. The antibody signals were detected by ECL Western blotting substrate (BIO-RAD) and blots were visualized using the BIO-RAD Gel documentation system (ChemiDoc MP System).

### 2.13. Histopathology

Morphological changes in testicular tissues from each group were examined under light microscopy. Tissue portion were treated in fixative solution [85 mL absolute alcohol + 5 ml glacial acetic acid + 10 mL formaldehyde (40%)]. After dehydration, tissue portions were embedded in paraffin blocks. Tissue blocks were sectioned 4–5 μm with a microtome and stained with Hematoxylin-Eosin (H&E) and studied under a light microscope (DIALUX 20 EB) at 20X.

### 2.14. Morphometry and Planimetry

For morphometric studies, the seminiferous tubule diameter and seminiferous tubule epithelial height of testicular tissue were measured by using Image J software (National Institute of Health, Bethesda, MD, USA). An image of the calibrated size in a micrometer was used for standard scale and conversion of values from pixels to micrometers. The area of the seminiferous tubule and interstitial space was calculated by planimetry by Image J software. The area in μm^2^ was measured using the method described previously [25]. Concisely, 25 pictures/animal (40×) of the identified area were selected and the area of seminiferous tubules and interstitial space was measured by the free selection tool of the software. The area percentage (%) was attained by the formula:

% Area of seminiferous tubule (AS) = [$\frac{\mathrm{As}\;x\;100}{T}$], where T is the total area of the field.

The percentage of the mean area was analyzed for a comparison between the treated and control groups.

### 2.15. Molecular Docking

Protein structures of TNF-α, STAT3, COX-3, NF-κβ 3β-HSD, CatSper, CYP20A1, and StAR were searched in RCSB Protein Data Bank [46]. After ensuring its unavailability, the ab initio predicted structures of these proteins were retrieved from the AlphaFold database [47]. AlphaFold ID for the tertiary structures is TNF-α AF-P0137, NF-κβ AF-Q04206, STAT3 AF-P40763, COX-3 AF- Q9Y6N1, 3β-HSD AF-Q9H2F3, CatSper AF-Q9NTU4, CYP20A1 AF-Q6UW02, and StAR AF-P49675. A two-dimensional structure for CDDP (CID: 5702198) was obtained from the PubChem database [48]. MMINA structure was drawn on ChemDraw 2016 and energy was minimized for obtaining the best suitable atomic conformation of the compound using a minimization plugin Chem3D [49]. Molecular docking between ligand and protein was performed through a web-based server CB dock [50]. CB dock employs a rigid docking approach and presents binding energy as a vina score. The selection of the best-docked model was based on the cavity size and vina score.

### 2.16. Pathway Construction

The construction of molecular interaction between TNF-α, STAT3, COX-3, and NF-κβ as well as StAR, CYP20A, 3β-HSD, and CatSper was performed through KEGG [51], DAVID [52], and STRING [53] databases. KEGG pathway maps map04630, map04920, map04923, map04927, and map00140 were used for understanding the protein–protein interactions. The cellular pathway determined was then drawn using the vector drawing program Inkscape ver 1.2 [54].

### 2.17. Statistics

Statistical analysis was done using GraphPad Prism version 9. Data were analyzed by one-way ANOVA following multiple comparison tests to study the statistical significance in different groups. The data obtained from qRT-PCR were evaluated using StepOne Software v2.2 RQ Study (Applied Biosystems/Life Technologies Corp., Carlsbad, CA, USA) and were exported as RQmax and RQmin (2-∆∆Ct +/− ∆∆Ct SD). These results represent the relative quantity of the gene of interest. Conversely, RQmax and RQmin represent the incorporation of the standard deviation of the ∆∆CT into the fold change calculations.

## 3. Results

### 3.1. Compound Synthesis

The starting material, indole hydrazide, 2-(5-methoxy-2-methyl-1H-indol-3-yl) acetohydrazide (1), was used for the synthesis of MMINA, 2-(5-methoxy-2-methyl-1H-indole-3-yl)-N’-[(E)-(3-nitrophenyl) methylidene] acetohydrazide. Appendix A illustrates the synthesis of the sulindac acetohydrazide derivative (MMINA) (Appendix A). The details of compound synthesis were published in our previous reports [31,32].

### 3.2. Testis and Epididymis Weight after Comparative Treatment Using MMINA and CDDP

Male rats were treated for 14 days in 5 different groups (*n* = 7 per group), as described in the Methods section. The groups included Control (saline), DMSO, CDDP, MMINA + CDDP, and MMINA. We observed no significant reduction in body weight in all treatment groups; however, noticeable changes were recorded in testis and epididymis weight (Figure 1). As a preliminary sign of organ impairment, organ weight is a clear indicator. In contrast to the Control and MMINA (25 mg/kg b.w.) groups, CDDP (12 mg/kg b.w.) injection induced a significant reduction in testis (*p* < 0.0001) and epididymis (*p* < 0.05) weight. MMINA co-treatment markedly attenuated the reduction of testicular and epididymis weight in comparison to the CDDP group at a statistical difference of *p* < 0.001 and *p* < 0.05, respectively.

### 3.3. Daily Sperm Production and Sperm Number in Various Parts of Epididymis

CDDP injection severely affected daily sperm production in comparison to control and MMINA (25 mg/kg b.w) treatment groups (*p* < 0.001). Similarly, there is an obvious (*p* < 0.0001) decline in the number of sperm in Cauda and Caput/Corpus epididymis in the CDDP-treated group. MMINA + CDDP administration rescued CDDP-induced loss in daily sperm production in the testis (*p* < 0.05) as well as sperm concentration in the various epididymis structures (*p* < 0.0001) (Table 1).

### 3.4. CDDP Induced Deregulation of Sex Hormones and Testosterone Biosynthesis Pathway

The effects of CDDP and/or MMINA on plasma testosterone and LH concentration are shown in Figure 2A(I). Significant reduction in the plasma testosterone and LH level was determined for the CDDP-treated group compared with the control group (*p* < 0.001). MMINA + CDDP treatment restored the plasma sex hormones to concentrations similar to control. StAR, CYP11A1, and 3β-HSD are the main steroidogenic enzymes that catalyze the biosynthesis of testosterone. Protein and gene expression studies indicated the negative regulation of these steroidogenic activators after CDDP injection. Following CDDP treatment, StAR, CYP11A1, and 3β-HSD mRNA expression were downregulated by 58.76% (*p* < 0.0001), 51.47% (*p* < 0.0001), and 45.8% (*p* < 0.001), respectively (Figure 2A(II),2B(II) and 2C(II). Expression of StAR, CYP11A1, and 3β-HSD proteins were significantly downregulated in CDDP-treated rat testis compared to control groups (Figure 2B(III). MMINA + CDDP treatment demonstrated significantly higher protein and mRNA expression of StAR and 3β-HSD compared to CDDP alone.

### 3.5. Sperm Motility and Epididymis CatSper Channel Activation

Epididymis sperm motility was significantly affected by CDDP exposure (*p* < 0.0001, Figure 3a). The MMINA + CDDP treatment group showed improved sperm motility in comparison to the CDDP alone treatment group (*p* < 0.001). Sperm motility is the main factor in fertilization capability. We assessed the effect of MMINA treatment on CatSper1 and CatSper2 channels. CatSper1 and CatSper2 are key regulatory channels in sperm hyperactivation. CatSper1 and CatSper2 mRNA expression levels were significantly reduced in the CDDP group in comparison to the control groups (*p* < 0.0001, Figure 3c,d); correspondingly, CatSper1 and CatSper2 protein abundances were also decreased due to CDDP treatment (Figure 3b) MMINA + CDDP treatment resulted in significantly increased CatSper 1 and 2 mRNA expression relative to CDDP alone (*p* < 0.0001, Figure 3c,d, respectively). MMINA treatment alone increased CatSper1 and 2 mRNA levels above those of control (*p* < 0.01 for CatSper1, Figure 3c,d). Consistent with RT-PCR results, immunoblot analysis validated that the expression of CatSper family proteins in the MMINA + CDDP-treated group was notably higher in comparison to the CDDP group.

### 3.6. MMINA Attenuated CDDP-Induced Oxidative Stress by Triggering Antioxidant Production

A total antioxidant capacity assay (TAC) was performed to measure the cellular antioxidant levels of the tissues. CDDP inoculation significantly weakens the total antioxidant capacity of the testis (*p* < 0.0001, Table 2). Furthermore, antioxidation enzymes comprising glutathione peroxidase (Gpx) and superoxide dismutase (SOD) activities were significantly lowered after CDDP treatment (*p* < 0.0001). Besides compromised antioxidant defense, the level of oxidative stress parameters, i.e., TBARs and NO, were also significantly elevated (*p* < 0.0001, Table 2). MMINA + CDDP treatment improved the total antioxidant capacity of testicular tissue relative to CDDP alone (*p* < 0.001). MMINA + CDDP treatment also significantly rescued depletion of Gpx and SOD concentration in comparison to CDDP alone (*p* < 0.0001), and partially restored TBARS and NO levels to near Control levels.

### 3.7. MMINA Inhibited Activation of Inflammatory Pathways in CDDP-Exposed Testicular Tissue

Stimulation of STAT3 and downstream activation of COX-2 and TNF-α are well-known signals for organ fibrosis. To examine whether MMINA is capable to moderate CDDP-induced testicular inflammation, we examined inflammatory indicators via RT-PCR and immunoblotting CDDP-induced upregulation of gene expression of STAT3, TNF-α, and COX-2 compared with the control group (*p* < 0.0001, Figure 4a). In parallel with mRNA expression data, the protein levels of these transcription factors in the CDDP administration group are significantly elevated in the testis in contrast to control. MMINA + CDDP treatment significantly inhibited the activation of STAT3, COX-2, and TNF-α at both mRNA and protein expression levels (Figure 4a,b).

### 3.8. MMINA Attenuated CDDP-Induced Drop in Spermatogenesis and Testicular Impairments

Histological examination was performed to study the effect of various treatments on tissue morphological features. Control, DMSO, and MMINA alone treatment groups demonstrated regular morphological features of the testis with intact spermatogenic layers. CDDP single-dose inoculation resulted in extreme cellular degeneration and disorganization in testicular cellular arrangements. There was an obvious loss of spermatogenic cells in the CDDP treatment group (Figure 5a). Morphometric scores reveal a significant decrease in seminiferous tubule diameter (STD, *p* < 0.001), seminiferous tubular area (STA % decrease, *p* < 0.0001), and seminiferous tubule epithelial height (STEH, *p* < 0.0001). A noteworthy diminution of percent sperm in the lumen (lumen with sperms, LS, *p* < 0.0001) was observed. Besides testicular cellular morphological deteriorations, the number of sperm was severely reduced in the CDDP group, and a significant increase in interstitial space (IS, percentage, *p* < 0.0001) and tubular lumen (TL, *p* < 0.0001) without sperm was recorded (Figure 5b,c). MMINA administration with CDDP rescued testicular structure and significant recovery of spermatogenesis was recorded. An increase in spermatogenesis was observed in the MMINA + CDDP group relative to CDDP alone by histopathological observation.

### 3.9. Molecular Interaction of TNF-α, STAT3, COX-2, and NF-κβ with MMINA and Cisplatin

MMINA was molecularly docked with TNF-α, STAT3, COX-2, and NF-κβ proteins through a rigid docking approach. The analysis revealed that MMINA binds strongly with all four proteins, whereas CDDP did not show any interaction with any of these genes. Among all the molecular complexes, molecular interactome analysis revealed that MMINA made four hydrophobic interactions with COX-2 His385, Tyr384, His387, and His206 (Table 3 and Appendix A). NF-κβ residue Lys37 made one hydrogen bond with MMINA, whereas MMINA made two hydrogen bonds with Lys244 and Thr456 of STAT3. TNF-αresidue His91 formed a hydrogen bond, while Val93 and Asn110 made hydrophobic interactions with TNF-α (Figure 6).

### 3.10. MMINA Modulated Molecular Interactions in Reducing Testicular Inflammation

Expression analysis revealed that MMINA along with CDDP reduced the protein and mRNA expression of COX-2, STAT3, TNF-α, and NF-κβ. Pathway analysis revealed that dimerized STAT3 and NF-κβ induce the expression of pro-inflammatory genes. STAT3 dimerization is induced through JAK-STAT pathway activation [55], whereas TNF-α signaling mediates the degradation of Iκβ (inhibitor of NF-κβ), promoting NF-κβ activation [56,57]. Furthermore, insulin receptor activation initiates the COX-2-driven signaling cascade that also activates IKK (an essential upstream player of NF-κβ signaling) (KEGG: map04920) (Figure 7). MMINA activity decreases the expression of COX2, STAT3, TNF-α, and NF- κβ; hence, this activity mediates its antiinflammatory effects.

### 3.11. Molecular Interaction of 3β-HSD, CatSper, CYP20A1, and StAR with MMINA and CDDP

Molecular docking of MMINA with 3β-HSD, CatSper, CYP20A1, and StAR proteins was also performed. The analysis showed the strongest binding of MMINA with 3β-HSD (binding affinity −9.6) and StAR (binding affinity −9) in comparison to CatSper and CYP20A1 (Table 4, Appendix A). Molecular interactome outcomes depicted the formation of one hydrogen bond and one hydrophobic interaction with 3β-HSD that contributed to its strong binding with MMINA. CatSper also made one hydrophobic interaction and one hydrogen bond with MMINA, but with a lower binding affinity of -7.7 relative to 3β-HSD. Furthermore, the distance of hydrogen bond between MMINA and CatSper residue Phe52 was greater (3.21 A) than between MMINA and 3β-HSD residue Ile194 (3.09 A) (Figure 8).

### 3.12. MMINA Modulated Molecular Interactions in Promoting Testicular Efficiency

Pathway analysis indicated that cAMP/PKA signaling mediated transcription factors (SF1, Sp1, Plox1, CREB, and NGF1β) activation and their nuclear translocation. This group of transcription factors promotes StAR, CYP17A1, and 3βHSD gene expression. These genes then mediate the production of progesterone from cholesterol and testosterone from progesterone. Progesterone and testosterone are two essential hormones in ensuring testicular efficiency. Progesterone further mediates sperm motility by activating the sperm-specific calcium ion channel CatSper via ABHD14/2. The production of these hormones is regulated by the action of StAR, CYP17A1, and 3β-HSD (Figure 9). Calcium ions (Ca^2+^) play a significant role in inducing the expression of these enzymes by bringing about the activation of transcription factors SF1, Sp1, Plox1, CREB, and NGF1β (KEGG: map04927) [58]. MMINA contributes by activating CatSper and StAR, CYP17A1, and 3βHSD gene expression. Pathway analysis speculated that MMINA might act by regulating or interacting with various upstream transcription factors to govern a positive impact on testicular functions. Furthermore, MMINA also promotes the activity of the Ca2+ ion channel CatSper and contributes to increasing sperm motility.

## 4. Discussion

Patients undergoing CDDP chemotherapy frequently experience fertility disorders. The mechanism(s) for testicular impairment after CDDP inoculation is still under investigation. Other platinum-holding drugs (e.g., carboplatin and oxaliplatin) have been utilized, but they are less effective and possess various adverse effects, such as myelosuppression and neuron and vestibular system damage [59]. In contrast, the cure rate of patients undergoing CDDP therapy with malignant lymphomas and germ cell tumors is 80 to 90%, respectively. Therefore, it is important to explore the possibilities to advance the clinical safety profile of CDDP.

Besides CDDP side effects, malignancy itself strongly disturbs spermatogenesis; therefore, explorations are underway for the identification of bioactive and less cytotoxic metabolites. Indoles have attracted a great deal of attention amongst scientists due to their therapeutic potential. Indole core has been continuously attracting the attention of researchers and designated as “privileged scaffolds”, which bind to multiple receptors with high affinity, leading to the development of target-based design and anticancer agents [28]. Indole derivative ‘MMINA’ possesses various pharmacological potentials and provides efficient protection of side effects induced by CDDP in various organs [31]. Hence, we investigated the antiinflammatory and therapeutic potential of indole acetohydrazide derivative, MMINA, against CDDP-induced testicular impairments.

CDDP exposure increases risk factors, such as stimulation of ROS production, downregulation of steroidogenic pathways, and subsequent defects in sperm functions. Therefore, consideration of genes regulating steroid hormone synthesis, sperm activation, and motility is very essential. The rodent model of CDDP-induced toxicity allows studying these factors to mimic the influence on human reproduction.

In the current investigation, we study the effect of CDDP exposure on testosterone and LH secretion in serum, expression levels of StAR, CYP20A1, 3β-HSD, and inflammatory biomarkers, spermatogenesis, histopathology, and redox balance in testis. Moreover, expression of CatSper genes and sperm motility were tested in the epididymis. A single treatment of CDDP (12mg/kg b.w.) did not induce any significant weight loss; however, a noticeable reduction in testis and epididymis weight was recorded. Testis weight is influenced by the number of differentiated germ cells, and its morphological and functional balance is driven by the sufficient production of male reproductive hormones. The reduction of testicular weight is correlated with poor spermatogenesis and steroid hormone synthesis [60]. MMINA administration along with CDDP rescued the weight reduction in both reproductive organs. In the current investigation, CDDP administration significantly altered male sex hormones testosterone and LH levels in plasma. Comparable consequences were reported by the Adejuwon group, indicating that CDDP administration dropped the reproductive hormones, linking altered Leydig cell function and disturbed spermatogenesis as a reason for low hormone synthesis in experimental models [61]. MMINA treatment significantly raised both testosterone and LH levels, indicating the modulatory effect of MMINA on the hypothalamic-pituitary-gonadal (HPG) axis as CDDP perturbations of sex hormones are linked to the dysregulated activity of the HPG axis [62].

We examined the effect of MMINA on steroidogenic regulatory pathways. StAR is the key factor in the steroid biosynthesis pathway, by transporting cholesterol to the inner mitochondrial membrane. In Leydig cells, StAR expression is regulated by LH secretion. Therefore, low expression of StAR crosslinks with testicular impairments. Equally, 3β-HSD is involved in the production of androstenedione in mitochondria and continues the testosterone biosynthesis. CYP20A1 is rate-limiting in steroidogenesis and is detected specifically in Leydig cells of the testis. Hence, downregulation of both StAR and 3β-HSD indicated poor secretion of testosterone [38,63]. Our data indicated that CDDP administration adversely affects testis through the poor production of testosterone. CDDP caused downregulation of StAR, CYP20A1, and 3β-HSD genes, leading to a decline in testosterone biosynthesis. CDDP-induced reduction in hormonal levels may be explained by the serious damage to Leydig and Sertoli cells as one of the possible mechanisms, resulting from the increased generation of free radicals [10]. MMINA co-treatment with CDDP restored the mRNA and well as protein levels of StAR, CYP20A1, and 3βHSD in testicular tissue. Our data indicated that MMINA increased testosterone concentrations by rescuing suppression of the testosterone biosynthesis pathway in rats.

We observed an increase in TBARs and NO levels in testicular tissue of the CDDP treatment group. Our results accord with those published earlier [10]. The mechanism of protection is via modulating the oxidative stress in testicular tissues; namely, MMINA treatment upregulated antioxidant (Gpx and SOD) biomarkers in the testis. Likewise, Garcia et al. (2012) revealed that CDDP increased ROS in Leydig cells of testis while downregulating the mRNA expression of cytochrome P450, resulting in testosterone synthesis ailments in Leydig cells of the mouse [64]. Oxidative stress is an intrinsic mechanism underlying testicular injury in CDDP-exposed rats [65]. Similar to the previous work of Zhao and colleagues, our data showed that CDDP decreases the activities of antioxidant biomarkers, such as SOD, GSH-Px, and TAC, while triggering the overproduction of MDA and NO as a critical indicator of oxidative trauma, validated testicular toxicity via eliciting oxidative stress in rats. MMINA combined administration with CDDP effectively rescued testicular cells from oxidative trauma and balanced the redox state of testicular tissues, confirming its antioxidant activity.

Besides spermatogenesis, sperm motility is an important factor in fertilization. During fertilization, sperm require hyperactivation to penetrate the oocyte zona pellucida. Motility stimulation is a prerequisite after sperm are released from the cauda epididymis. Intracellular Ca2+ levels were shown to regulate sperm motility and hyperactivation, capacitation, and the acrosome reaction. In this study, sperm motility and morphology were negatively affected by CDDP treatment with reversal of these effects due to MMINA treatment. We recorded a significant increase and speculated that this surge in DSP was cross-linked with MMINA antioxidant effects. The epididymis is the main regulatory tissue that governs sperm storage and transportation and is important in sperm motility regulation as well. Our data correlate with previous investigations indicating that a single dose of CDDP (10 mg/kg) in rats caused reduced sperm motility, sperm density, sperm count, and an associated decline in testosterone, LH, and FSH levels in serum [66]. As CatSper 1 and 2 are specifically expressed in reproductive organs and key regulatory factors in sperm motility, we examined the expression of CatSper channel isoforms 1. We observed a CDDP-induced reduction in daily sperm number and sperm motility in the MMINA + CDDP treatment group. CDDP may cause immobility in sperm by blocking CatSper-specific calcium channels. MMINA upregulated CatSper-specific calcium channels, reversing the CDDP effects.

STAT3 and NF-kB are inflammation-associated transcription factors that regulate many pro-inflammatory cytokines, such as TNF-α and COX-2 [67]. COX-2 is typically not expressed in normal human testes; however, its expression is induced in rat Leydig cells during impaired spermatogenesis and is a responsible factor in the decrease of testosterone production by testis [68]. CDDP exposure triggered inflammatory cells, and consequently, magnified the inflammatory response by releasing various cytokines, such as TNF-α and COX-2, resulting in damage to the testes [69]. Indoles are non-steroidal antiinflammatory drugs (NSAIDs) that produce their activity via inhibition of cyclooxygenase (COX) catalyzed biotransformation of arachidonic acid to pro-inflammatory prostaglandins (PGs) and thromboxanes. The COX-2 isoform is induced in response to mitogenic and pro-inflammatory stimuli and it is responsible for the progression of inflammation [70]. MMINA inhibits the expression of COX2 and exerts its antiinflammatory response (Figure 9). Similarly, selective inhibition of COX2 has been observed previously in the case of celecoxib, rofecoxib, and valdecoxib [28]. These selective COX-2 inhibitor drugs were more useful for the treatment of inflammation and inflammation-associated disorders.

We observed that the treatment of rats with CDDP stimulated the expression of STAT3, NF-*κ*B, TNF-α, and COX-2 at both mRNA and protein levels. MMINA co-treatment inhibited STAT3 gene expression as well as phosphorylation of pSTAT3 (Y705) protein and downregulated the expression of pro-inflammatory response genes. Considering that TNF-α is one of the cytokines driven by STAT3 signaling [25,26], we could speculate that the effect of TNF-α on COX-2 is predicted to be via pSTAT3 (Y705)/NF-κB p65-associated mechanisms. In general, inflammation-associated factors could reciprocally activate each other to form a complex network to further magnify the inflammatory response [71]. Therefore, the inactivation of pSTAT3 (Y705), NF-кB p65 signaling, and associated downregulation of COX-2 and TNF-α expression in testis might be the mechanism underlying the antiinflammatory response generated by MMINA after CDDP inoculation. MMINA showed protection against CDDP-induced broad-spectrum organ damage. Modifications to the indole are efficient in protecting against chemotherapeutic drug-induced side effects, as the low toxicity of MMINA highlights the translational value of MMINA in cancer prevention/therapy. These results indicate that it may be a useful lead for anticancer drug development in the future [72].

The histopathological analysis depicts a high magnitude of damage in testicular tissues. A previous report indicated that CDDP (10 mg/kg) could cause irregular seminiferous tubules, and a reduction of seminiferous epithelial layers and germ cells in rats [73]. Similar findings were observed in the current investigation using a 12 mg/kg dose of CDDP. We observed disorganized seminiferous epithelium, impaired spermatogenic cells, empty seminiferous tubule lumen, and thin seminiferous epithelium. Morphometric analysis indicated that CDDP significantly altered testicular morphology with an increase in diameter and area of seminiferous tubule lumen. We noticed that testicular histopathological changes were improved in CDDP when combined with MMINA. The above results indicated that the rat model of testicular lesion induced by CDDP was successfully established and MMINA had protective effects on CDDP-induced testicular dysfunction.

## 5. Conclusions

The current investigation provided multiple levels of evidence that CDDP exposure prompted lethal effects on testis via triggering oxidative stress/inflammation and morphological alterations, compromising antioxidant defense mechanism, and inhibiting testosterone synthesis in a rodent model of toxicity. The indole derivative “MMINA” demonstrated significant protective outcomes against CDDP-induced testicular damages in vivo. The improvement of the testicular structure and sperm production in the MMINA co-treatment group has likely resulted from the following combined effects: (1) reduced oxidative stress and STAT3, COX-2, NF-кB, and TNF-α associated stress and inflammatory response in the testis; (2) increased total antioxidant capacity of testicular tissues and reproductive hormones especially testosterone biosynthesis that is linked with an increase in spermatogenesis; (3) enhancement of StAR, 3βHSD, and CPY20A1 expression in testicular tissue resulting in improved testicular function and restoration of normal sperm production; (4) improved sperm motility and hyperactivation via induction of CatSper expression that is prerequisite for the fertilization. An increase in calcium channel components CatSper1 and 2 signals the beneficial potential of MMINA to treat reproductive dysfunction and male infertility. This study provides evidence that an increase in CatSper, StAR, 3β-HSD, and CYP20A1 and downregulation of STAT3, COX-2, and associated signaling pathways can provide an effective strategy to prevent CDDP-induced reproductive toxicity and infertility disorders. Our findings render opportunities for re-introducing CDDP to systemic anticancer therapy with significantly reduced testicular toxicity, and our results would potentially be valuable in chemotherapy-related infertility.

## Figures and Tables

**Figure 1 antioxidants-11-02063-f001:**
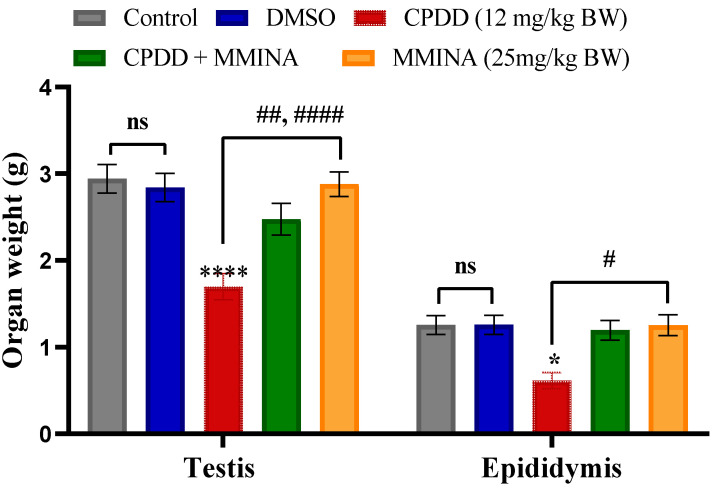
Effect of various treatments on organ weights. *, **** *p* < 0.05 and *p* < 0.0001 versus Control respectively, ^#,##,^
^####^
*p* < 0.05, *p* < 0.01 add *p* < 0.0001 versus CPDD. ns = non-significant.

**Figure 2 antioxidants-11-02063-f002:**
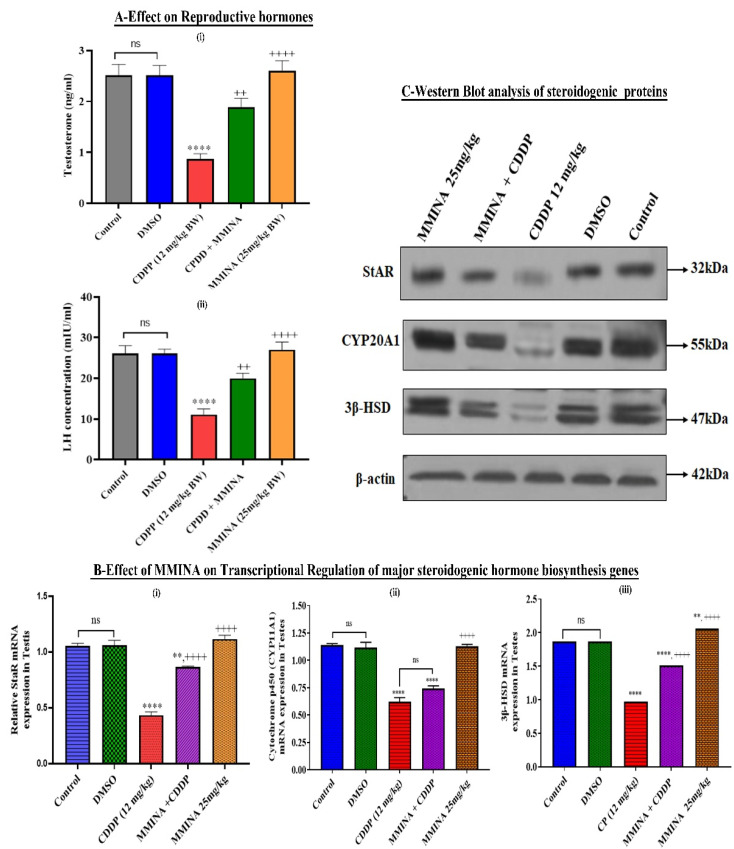
MMINA upregulated protein expression levels of the key testosterone synthesis factors and related hormones in rat testes. (**A**) Effect on reproductive hormones; (i): testosterone, (ii): LH. (**B**) Effect of MMINA on transcriptional regulation of the steroidogenic hormone biosynthesis gene. rtPCR analysis of StAR (i), CYP11A1 (ii), and 3β-HSD (iii) mRNA expression. (**C**)-StAR, CYP11A1, and 3β-HSD protein content were determined by Western blot analysis. Values were expressed as mean ± standard deviation (*n* = 7). **, **** *p* < 0.05 and *p* < 0.0001 versus control, respectively, and ^++^, ^++++^
*p* < 0.05 and *p* < 0.0001 versus CDDP. Data were analyzed by one-way ANOVA followed by Tukey’s multiple comparison tests using graph pad prism version 9. STAR: steroidogenic acute regulatory protein, CYP11A1: P450 side-chain cleavage enzyme, HSD17B: 17β-hydroxysteroid dehydrogenase. Uncropped blots are presented in Appendix A.

**Figure 3 antioxidants-11-02063-f003:**
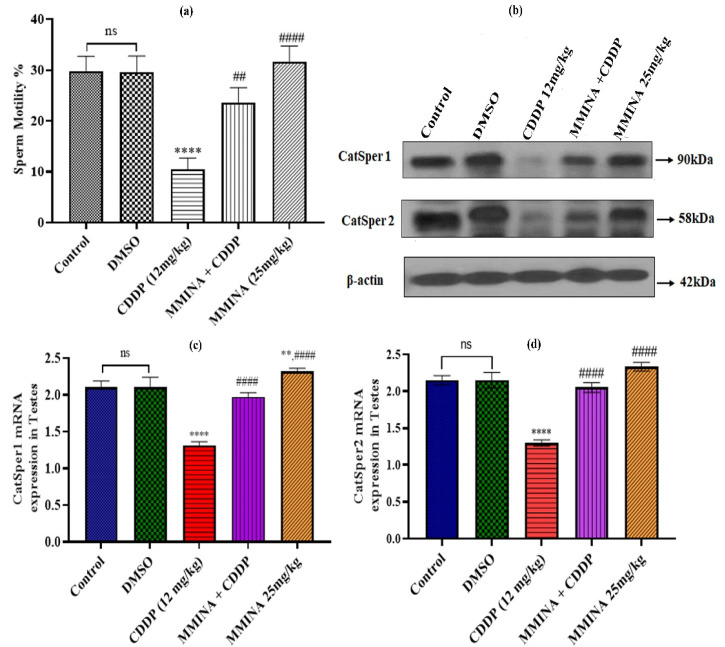
MMINA treatment enhanced sperm motility and activate the CatSper channel. (**a**): The percentage of motile spermatozoa in different treatment groups. (**b**): Western blot analysis indicating the protein expression of CatSper 1 and CatSper 2 channels in various treatment groups. MMINA administration prominently upregulates the expression of both channels in comparison to the CDDP alone exposed group. (**c**,**d**) Indicated the mRNA expression of CatSper 1 and 2 genes. Data are mean ± SEM, (*n* = 7). **, **** indicated significant variation at *p* < 0.01 and *p* < 0.0001, respectively, versus Control group and ^##^, ^####^ indicated significant variation at *p* < 0.05 and *p* < 0.0001, respectively, versus CDDP group. ns= non-significant. Uncropped blots are presented in Appendix A.

**Figure 4 antioxidants-11-02063-f004:**
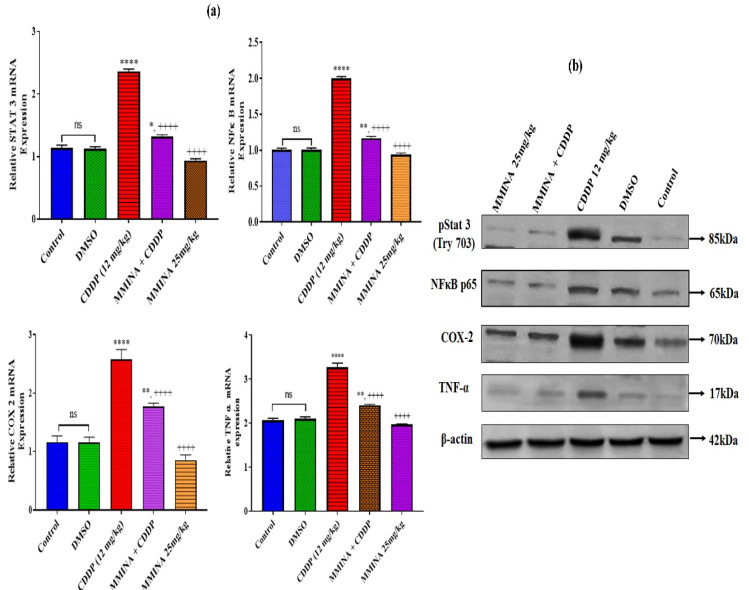
Comparison of fold change expression of target proteins and genes in various treatment groups. (**a**) RT-PCR analysis to determine the mRNA expression of STAT 3, NFκb, COX 2, and TNF-α. The error bars indicated the standard error of the mean expression level. Data are mean ± SEM (*n* = 7). *,**, **** indicated significant variation at *p* < 0.05, *p* < 0.01 and *p* < 0.0001 versus Control group, respectively, and ++++ indicated significant variation at *p* < 0.0001 versus CDDP group. (**b**) Protein expression analysis by Western blotting: β-actin was used as an endogenous control for the assessment of protein loading. Uncropped blots are presented in Appendix A.

**Figure 5 antioxidants-11-02063-f005:**
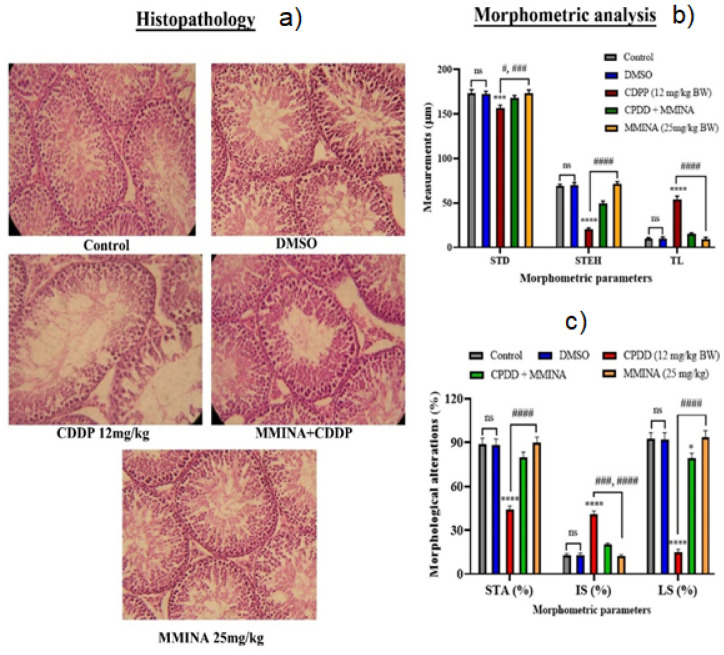
Light micrographs of testicular tissues of rats treated with MMINA and CDDP. Sections were stained with hematoxylin and eosin (40x). (**a**) Photomicrograph of the testicular tissues from various treatment groups. Control group showing typical seminiferous tubules structure at all stages of spermatogenic and the interstitial cells with Leydig cells filling the space between the seminiferous tubules. The CDDP (12 mg/.kg b.w) treatment group showed degenerative alterations in spermatogenic cells and the detachment of the spermatogenic epithelium. With empty lumen, the MMINA + CDDP group indicated a significant protective effect of MMINA against CDDP-induced morphological alterations and increase spermatogenesis. MMINA alone group showed a healthy histological structure similar to the control group. (**b**) Morphometric analysis of testicular tissue alterations in experimental groups was performed using Image J software. STD: seminiferous tubule diameter, STEH: Seminiferous tubule epithelial height, TL: Tubular lumen, STA: Seminiferous tubular area, IS: i2terstitial space, LS: Lumen with sperms. Data are mean ± SEM, (*n* = 7). *** indicated significant variation at *p* < 0.0001, respectively, versus Control group, and #, ###, #### indicated significant variation at *p* < 0.05, *p* < 0.001, and *p* < 0.0001 versus the CDDP group, ns= non-significant, respectively. Data were analyzed by one-way ANOVA followed by Tukey’s multiple comparison tests.

**Figure 6 antioxidants-11-02063-f006:**
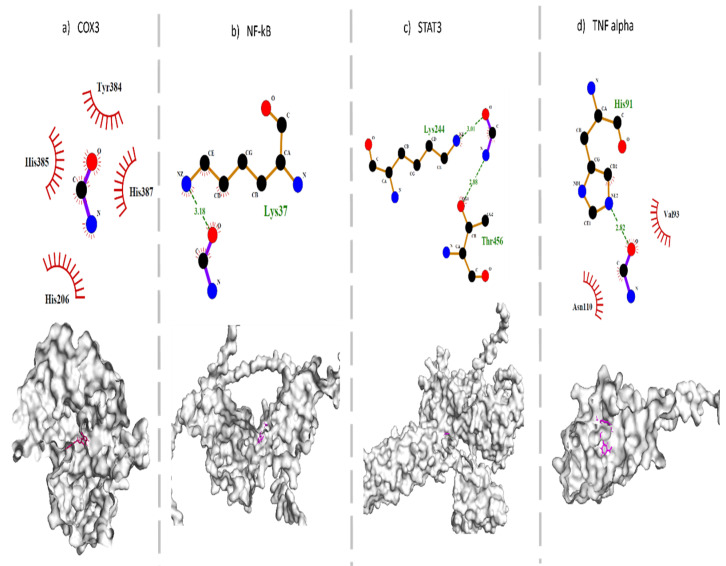
Molecular interactions between MMINA and pro-inflammatory proteins TNF-alpha, STAT3, COX-2, and NF-κβ**.** (**a**) The MMINA-COX2 complex has predominant hydrophobic interactions, whereas (**b**) the MMINA-NF-κβ complex and (**c**) the MMINA-STAT3 complex has hydrogen bonding, and (**d**) the MMINA-TNFalpha complex has both hydrogen bonding and hydrophobic interactions. Semi-circles in red color depict hydrophobic interactions, green dotted lines show hydrogen bonding, and green numbers indicate the distance between hydrogen-bonded ligand and amino acid. Purple-lined structure denotes ligand, while orange line structures represent amino acids. The surface view of the protein and ligand complex is shown below the 2D figure. Ligand is indicated with purple color.

**Figure 7 antioxidants-11-02063-f007:**
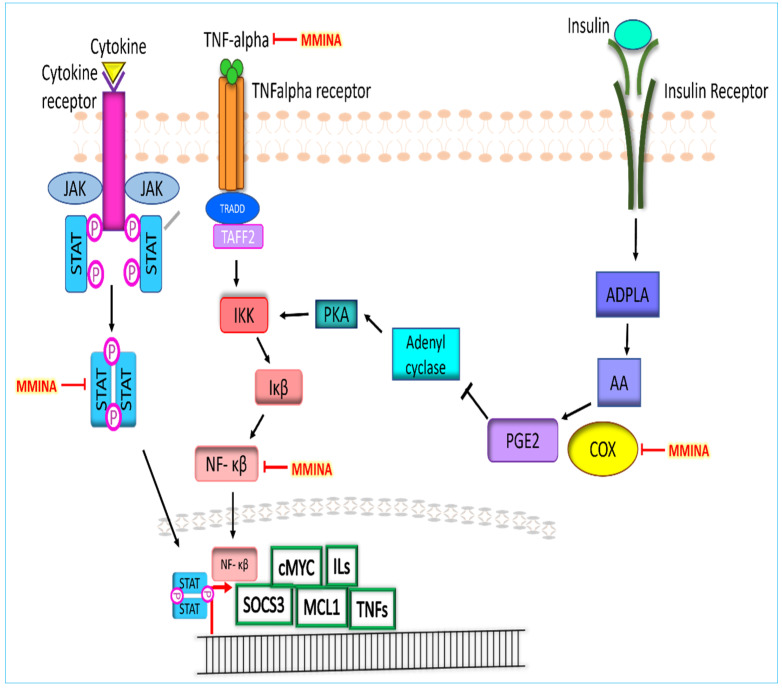
Proposed pathways of MMINA effect on the signal transduction of inflammatory biomarkers. Cytokine/JAK-STAT signaling and TNFalpha/NF-κβ signaling mainly contribute to the manifestation of inflammation. MMINA downregulates the expression of NF-κβ, TNFalpha, and STAT3 at mRNA and protein levels and reduces inflammation. COX2 pathway crosstalk also activates the NF-κβ signaling and contributes to inflammation. MMINA also regulates COX2 expression and inhibits this pathway.

**Figure 8 antioxidants-11-02063-f008:**
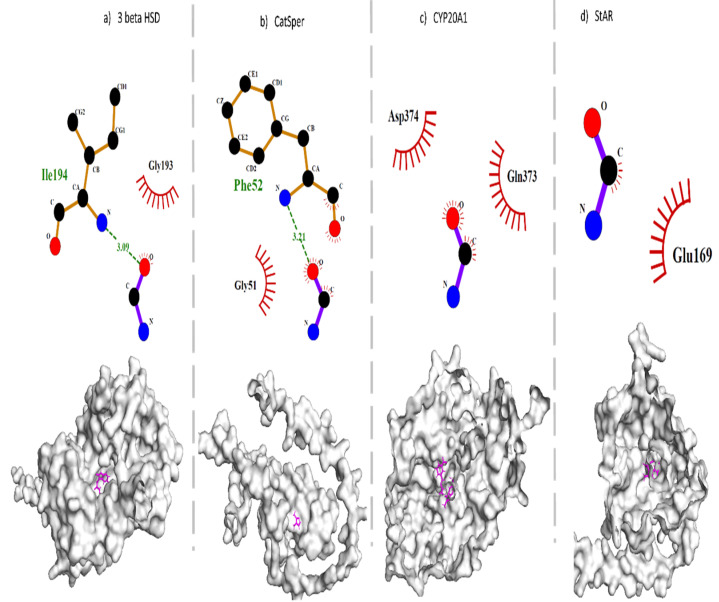
Molecular interactions between MMINA and 3β-HSD, CatSper, CYP20A1, and StAR. (**a**) MMINA-3β-HSD and (**b**) MMINA-CatSper complex has one hydrophobic interaction and one hydrogen bond, whereas hydrophobic interactions predominated in (**c**) MMINA-CYP20A1 and (**d**) MMINA-StAR complex. Semi-circles in red color depict hydrophobic interactions, green dotted lines show hydrogen bonding, and green numbers indicate the distance between hydrogen-bonded ligand and amino acid. Purple lined structure denotes ligand, while orange line structures represent amino acids. The surface view of the protein and ligand complex is shown below the 2D figure. Ligand is indicated with purple color.

**Figure 9 antioxidants-11-02063-f009:**
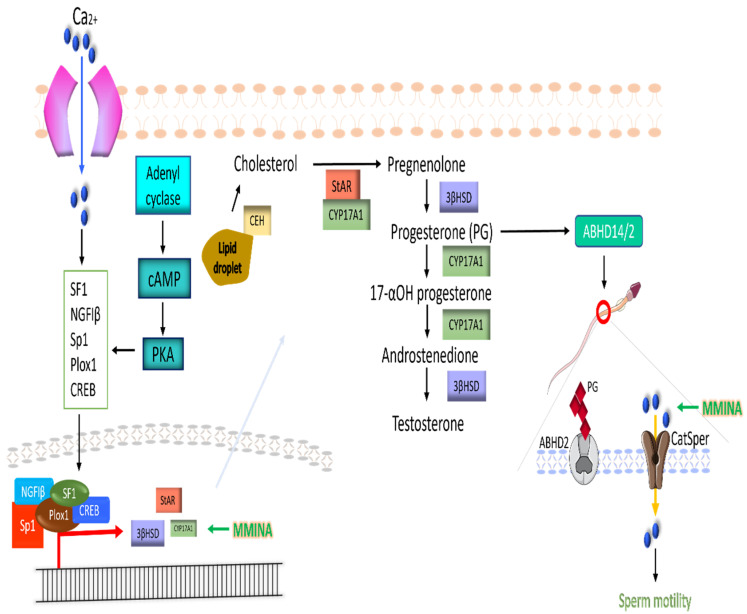
Proposed pathways of MMINA mediated expression of StAR, CYP17A1, 3βHSD, and CatSper in increasing testicular efficiency. StAR, CYP17A1, and 3βHSD are modulators of progesterone and testosterone production. Their expression is mediated by calcium ions and cAMP/PKA signaling mediated transcription factors (SF1, Sp1, Plox1, CREB, and NGF1β) activation and their nuclear translocation. This group of transcription factors promotes StAR, CYP17A1, and 3βHSD gene expression. These genes then mediate the production of progesterone from cholesterol and testosterone from progesterone. Progesterone further mediates sperm motility by activating the sperm-specific calcium ion channel CatSper via ABHD14/2. MMINA contributes by activating CatSper and StAR, CYP17A1, and 3βHSD gene expression.

**Table 1 antioxidants-11-02063-t001:** Effect of MMINA treatment on daily sperm production (DSP) and sperm number in the epididymis.

TREATMENTS	DAILY SPERM PRODUCTION (×106)	CAPUT/CORPUS EPIDIDYMIS SPERM NUMBER (×106/G ORGAN)	CAUDA EPIDIDYMIS SPERM NUMBER (×106/G ORGAN)
**CONTROL**	50.65 ± 1.11	315.58 ± 5.53	546.35 ± 7.06
**DMSO VEHICLE**	50.71 ± 1.20	316.01 ± 5.82	546.90 ± 7.13
**CDDP (12 MG/KG)**	27.50 ± 0.98 **	138.18 ± 3.83 ****	217.17 ± 4.45 ****
**MMINA + CDDP**	47.54 ± 1.01 ^+^	291.17 ± 4.17 **, ^++++^	520.31 ± 5.97 **, ^++++^
**MMINA (25 MG/KG)**	51.37 ± 1.14 ^++^	319.01 ± 5.91 ^++++^	550.76 ± 6.81 ^++++^

Data are mean ± SEM, (*n* = 7). **, **** *p* < 0.01 and *p* < 0.0001 versus Control, respectively, and ^+^,^++^,^++++^
*p* < 0.05, *p* < 0.001, *p* < 0.0001, versus CDDP. Data were analyzed by one-way ANOVA followed by Tukey’s multiple comparison tests.

**Table 2 antioxidants-11-02063-t002:** Effect of various treatments on testicular redox status and total antioxidant capacity.

GROUPS	TBARS (NG/MG)	NO (NG/MG)	SOD (U/G)	GPX (NM/MIN/G)	TAC (URIC ACID EQUIVALENT MM)
**CONTROL**	43.51 ± 2.16	21.13 ± 1.01	49.67 ±1.20	58.25 ± 1.20	0.51 ± 0.024
**DMSO VEHICLE**	43.98 ± 2.01	21.08 ± 1.14	49.65 ± 1.09	58.17 ± 1.19	0.51 ± 0.032
**CDDP (12 MG/KG)**	151.50 ± 3.46 ****	98.54 ± 2.41 ****	6.01 ± 0.41 ****	6.75 ± 0.56 ****	0.30 ± 0.022 ****
**CDDP + MMINA**	57.67 ± 2.81 *,^++++^	32.69± 2.11 *,^++++^	39.99 ± 1.01 ^++++^	46.01 ±1.05 ^++++^	0.44 ± 0.027 *,^+++^
**MMINA (25 MG/KG)**	42.02 ± 2.37 ^++++^	20.92 ± 1.07 ^++++^	48.69 ± 1.23 ^++++^	59.35 ± 1.11 ^++++^	0.53 ± 0.041 ^++++^

Data are mean ± SEM, (n = 7). *, **** *p* < 0.05 and *p* < 0.0001 versus Control, respectively, and ^+++^, ^++++^
*p* < 0.001 and *p* < 0.0001 versus CDDP, respectively. Data were analyzed by one-way ANOVA followed by Tukey’s multiple comparison tests.

**Table 3 antioxidants-11-02063-t003:** MMINA and inflammatory pathway proteins docked complexes along with the vina score.

DRUG	PROTEIN	VINA SCORE
**MMINA**	COX-2	−9.1
	TNF-α	−7.6
	NF-κB	−7.3
	STAT3	−8.6

**Table 4 antioxidants-11-02063-t004:** MMINA and 3β-HSD, CatSper, CYP20A1, and StAR docked complexes along with the vina score.

Drug	Protein	Vina Score
**MMINA**	3β-HSD	−9.6
	CatSper	−7.7
	CYP20A1	−7.6
	StAR	−9

## Data Availability

All the data are contained in the manuscript.

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
