# Peer review of "Prevention of Testicular Damage by Indole Derivative MMINA via Upregulated StAR and CatSper Channels with Coincident Suppression of Oxidative Stress and Inflammation: In Silico and In Vivo Validation"

_antioxidants, 2022, doi:10.3390/antiox11102063_

Round 1
Reviewer 1 Report
The topic of the work is of interest; however, there are some issues and some crucial points that need consideration. Moreover, some grammar, editing and typing errors must be corrected. For that reasons, I suggest major revision of the manuscript before to assess it for publication in Antioxidants. In particular:
Pages 1-2, lines 18-48: Please shorten the abstract by leaving only the most important findings of the work.
Page 1, line 36: abbreviations should be fully explain only at the first mention
Page 4, line 148: The Authors should cite the specific OECD guideline used to determine the acute toxicity. Moreover, they should reported the category in which the LD50 falls.
Page 4, lines 163-168: the Authors should explain why they have chosen the intraperitoneal administration route instead of intravenous, being the latter the common way by which cisplatin is administered. Moreover, why did you administered MMINA i.p. and not orally, being a derivative of sulindac (NSAIDs)?
Page 4, lines 182-183: Please, replace the latin word “carpus” with “corpus”
Page 5, lines 219: Please, move Table 1 in the “Results” section at page 8 after line 343.
Page 7, lines 303-315: This paragraph is the same presented in the work 39. Please refer to it without repeating it.
Page 7, lines 316-323: This paragraph is also reported in the supplementary files. Please choose where do you want to display it.
Page 8, lines 352-353: The mRNA expression of StAR and 3β-HSD is reported in "Figure 2, IIa and IIc", and not in "Figure 2, IIa and IIb". Please correct the mistake. Why the Authors did not mention CYP11A1 mRNA expression (Figure 2, IIb)?
Page 9, Figure 2: Please organize the figure in a clearer way by indicating the images with different letters without numbers.
Page 11, lines 396-397: The authors said “…superoxide dismutase (SOD) activities were significantly down-regulated…”, maybe it is better to say that ““…superoxide dismutase (SOD) activities were significantly lowered…”
Page 12, lines 421-422: The authors said “The error bars show the min/max expression levels that measure the standard error of the mean expression level.”. Please rephrase the sentence in a clearer way.
Page 17, “Discussion” section: Please discuss your results by taking into account literature evidence. Are there other compounds tested as chemoprotective agent of testicular damage? What is the advantage, if any, of MMINA?
Pages 20-25, “Reference” section: Please, accurately revise all the references. Sometimes, the journal name is lacking (e.g., see reference 39).
Author Response
Dear Editor
We have addressed all the comments raised by the reviewer. The following are the answers.
Pages 1-2, lines 18-48: Please shorten the abstract by leaving only the most important findings of the work.
Response: The abstract section is shortened
Page 1, line 36: abbreviations should be fully explain only at the first mention
Response: full form added
Page 4, line 148: The Authors should cite the specific OECD guideline used to determine the acute toxicity. Moreover, they should reported the category in which the LD50 falls.
Respone: added
Page 4, lines 163-168: the Authors should explain why they have chosen the intraperitoneal administration route instead of intravenous, being the latter the common way by which cisplatin is administered. Moreover, why did you administered MMINA i.p. and not orally, being a derivative of sulindac (NSAIDs)?
Response: In rodent’s cisplatin is usually injected intraperitoneally (ip) and less frequently intravenously (iv) or subcutaneously (sc). There are wide number of publish researches reporting Ip administration route of Cisplatin. Intraperitoneal (IP) route of drug administration in laboratory animals is a common practice in many in vivo studies of disease models. As this route is an easy to master, quick, suitable with low impact of stress on laboratory rodents. IP administration of drugs in experimental studies involving rodents is a justifiable route for pharmacological and proof-of-concept studies where the goal is to evaluate the effect(s) drug.
Page 4, lines 182-183: Please, replace the latin word “carpus” with “corpus”
Response: corrected
Page 5, lines 219: Please, move Table 1 in the “Results” section at page 8 after line 343.
Response: corrected
Page 7, lines 303-315: This paragraph is the same presented in the work 39. Please refer to it without repeating it.
Response: removed as per your suggestion
Page 7, lines 316-323: This paragraph is also reported in the supplementary files. Please choose where you want to display it.
Response: removed from the main manuscript, only display in supplementary material.
Page 8, lines 352-353: The mRNA expression of StAR and 3β-HSD is reported in "Figure 2, IIa and IIc", and not in "Figure 2, IIa and IIb". Please correct the mistake. Why the Authors did not mention CYP11A1 mRNA expression (Figure 2, IIb)?
Response: corrected
Page 9, Figure 2: Please organize the figure in a clearer way by indicating the images with different letters without numbers.
Response: figure reorganized
Page 11, lines 396-397: The authors said “…superoxide dismutase (SOD) activities were significantly down-regulated…”, maybe it is better to say that ““…superoxide dismutase (SOD) activities were significantly lowered…”
Response: corrected as suggested
Page 12, lines 421-422: The authors said “The error bars show the min/max expression levels that measure the standard error of the mean expression level.”. Please rephrase the sentence in a clearer way.
Response: corrected
Page 17, “Discussion” section: Please discuss your results by taking into account literature evidence. Are there other compounds tested as chemoprotective agent of testicular damage? What is the advantage, if any, of MMINA?
Response:
Pages 20-25, “Reference” section: Please, accurately revise all the references. Sometimes, the journal name is lacking (e.g., see reference 39).
Response: corrected

Reviewer 2 Report
This is an interesting and comprehensive study on the effects of MMINA against CDDP-induced testicular toxicity in rats. The research is well conducted, and the results obtained are relevant from both scientific and clinical point of views.
The information given in the Introduction section is appropriate and the results are clearly exposed. The Discussion and the Conclusion sections agree with the research conducted and the results obtained.
I have only some minor comments to note:
1) The text has several typographical errors that must be corrected.
2) Discussion section the expression “our data indicated” is excessively used.
3) The quality of Figures 6, 7, 8 and 9 must be improved.
Author Response
Dear Editor
We have addressed all the comments raised by the reviewer. The following are the answers.
- The text has several typographical errors that must be corrected.
Response: Corrected
- Discussion section the expression “our data indicated” is excessively used.
Response: Reviewed
- The quality of Figures 6, 7, 8, and 9 must be improved.
Response: improved

Round 2
Reviewer 1 Report
Almost all my comments were addressed. Thus, I think that the manuscript can be accepted for publication in Antioxidants in present form.